# Increasing Health Literacy May Reduce Health Inequalities: Evidence from a National Population Survey in Ireland

**DOI:** 10.3390/ijerph17165891

**Published:** 2020-08-13

**Authors:** Sarah Gibney, Lucy Bruton, Catherine Ryan, Gerardine Doyle, Gillian Rowlands

**Affiliations:** 1Department of Health, Dublin 2, DO2 XW14, Ireland; sarah_gibney@health.gov.ie (S.G.); Lucy_Bruton@health.gov.ie (L.B.); catherineryan1997@gmail.com (C.R.); 2UCD College of Business, University College Dublin, Dublin 4, D04 V1W8, Ireland; gerardine.doyle@ucd.ie; 3UCD Geary Institute for Public Policy, University College Dublin, Dublin 4, D04 V1W8, Ireland; 4Population Health Sciences Institute, Newcastle University, Newcastle upon Tyne NE1 7RU, UK

**Keywords:** health literacy, health inequalities, health disparities, reduction of inequalities

## Abstract

*Background*. Health literacy has been separately associated with socio-economic status and worse health status and outcomes. However, the magnitude of the associations between health literacy and health status and outcomes may not be evenly distributed across society. This study aims to estimate and compare the associations between health status, health behaviours, and healthcare utilisation within different levels of social status in the Irish population. *Materials and methods*. Data from Ireland collected as part of the 2011 European Health Literacy Survey were analysed. General health literacy was measured on a 0–50 scale, low to high. There were four binary outcomes: long-standing health conditions, smoking, hospital visits in the last 12 months, and self-rated health status. Logistic regression analysis was conducted to estimate the likelihood of each health outcome. Health literacy was treated as the main independent variable. Marginal effects were calculated using the delta method to demonstrate the change in likelihood of each outcome associated with a 5-point increase in health literacy score. The sample was grouped into tertiles based on self-reported social status, and models were replicated and compared for each tertile. Models were adjusted for known correlates of health literacy and health: age, gender, and education. Analysis was conducted using Stata V14. *Results*. Higher health literacy scores were associated with a lower probability of having a limiting illness within the low social status group only. Higher health literacy scores were associated with a lower probability of three or more hospital visits in the past 12 months in the low and middle social status groups. For people in the low and middle social status groups, higher health literacy levels were associated with a lower probability of being a current smoker. The associations between health literacy and self-rated health status were similar in each social status group. *Conclusions*: Improvement in population health literacy may reduce the prevalence of long-term chronic health conditions, reduce smoking levels, and result in fewer hospital visits. Whilst improved health literacy should improve behaviours and outcomes in all groups, it should have a more marked impact in lower social status groups, and hence contribute to reducing the observed social disparities in these health outcomes.

## 1. Introduction

The term health inequality refers to the unjust nature of health differences between social groups, generated by social conditions, describing the pattern in which those from economically and socially poorer backgrounds run higher risks of premature death and contracting chronic or serious illness [1]. As such, health is costly to individuals, societies, and economies [2]. In parallel, health literacy is now well-established as a modifiable factor that contributes to the promotion and maintenance of good health and wellbeing across the life course. The importance of health literacy is increasingly recognised due to both the prevalence of low health literacy and the associations between low health literacy and poorer health, unhealthy behaviours, and higher rates of healthcare utilization [3]. Globally, there has been a sustained growth in efforts to measure and monitor health literacy at a population level, to understand how health literacy is distributed within populations, to understand the consequences of this distribution for population health, and ultimately to inform public health policies and strategies. Health literacy follows a socio-economic gradient [3], as do health outcomes [4] and health behaviours [5]. In order to better understand these interrelationships, this study aims to examine the impact of socio-economic position on the health literacy distribution of health status and outcomes in the Irish population. The goal of this study is to investigate the extent to which increasing health literacy may improve health outcomes and behaviours and to compare the effects of increasing health literacy at different levels of social status. It is intended that the results of this study will serve to inform a health-literacy approach to the design and monitoring of public health interventions in Ireland and elsewhere and may contribute to enhancing their efficiency and effectiveness.

Although life expectancy in Ireland has increased, there is evidence to suggest that people are not necessarily ageing in good health—noncommunicable diseases, such as hypertension and obesity, are increasingly common [6]. It has been estimated that approximately 5400 deaths per year in Ireland could be prevented if social inequalities in health were addressed and that this would involve detailed consideration of three influential factors: the distribution of income and government spending, psychosocial factors, such as stress and social support, and lifestyle or behavioural factors associated with different socio-economic statuses [1]. These factors interrelate and affect one another through the experience of poverty, leading to the substantial health gap between those of higher and lower socio-economic status [7]. The conditions of poverty can also contribute to worsened health, for example, chronic respiratory issues from living in damp housing or limited access to education and employment [1]. Addiction issues, such as smoking, alcohol, and drugs also disproportionally affect those from less affluent socio-economic backgrounds, exacerbating and creating additional chronic conditions such as high blood pressure and increasing the mortality rate due to smoking, drinking, and drug-related deaths. Mental health is also an issue, as there are higher rates of depression and hospitalisation among lower socio-economic classes in Ireland, often due to the mental health effects of poverty, systemic inequality, and material deprivation [8].

Analysis of health policy in Ireland suggests that the approach is attuned to the impact of poverty and inequality on health. However, limitations in current policies surrounding healthcare provision have been described as being underequipped to deal with more “unpredictable” factors, such as the onset of illness and disability [9], which place significant demands on individuals in terms of health literacy. Evidence from a European survey “Health Inequalities in Europe: Setting The Stage for Progressive Policy Action” [10] suggests that Ireland’s two-tiered healthcare system may also contribute to the gap of health inequity, insofar as a proportion of the population on low incomes are above the income threshold of entitlement for the General Medical Scheme (GMS), pay out-of-pocket for medical expenses, and cannot afford private medical insurance [10,11]. This report also noted that Ireland is one of the only countries in the European Union without universal healthcare coverage for all citizens [10], and this is now a key focus of the Sláintecare reform. It is clear that the health and social care system in Ireland is complex and, like many countries, also presents a particular challenge in terms of health literacy and public health, insofar as those most in need are also those who are likely to lack the health literacy skills to navigate the system and engage fully in programmes and interventions to improve health and wellbeing.

### 1.1. Health Literacy in Ireland

Health literacy is a broad concept. Functional health literacy refers to reading and writing capacities that assist in everyday health concerns, interactive health literacy refers to the ability to apply one’s knowledge of health and wellbeing to new circumstances, and critical health literacy refers to the ability to think critically and analyse health information objectively [12]. Low levels of health literacy can have serious negative health outcomes, such as an inability to identify and access information on illness, and to communicate about illness, conditions, or pain. Low levels of health literacy have been directly linked to higher rates of negative health outcomes, poor disease knowledge, low levels of preventive health service utilisation, lower levels of mental wellbeing, low medication adherence and earlier death, and poorer healthcare interactions [13,14].

The European Health Literacy Survey (HLS-EU), which took place in eight EU countries including Ireland, showed that the mean overall prevalence of inadequate or problematic health literacy was 47.6%, with Ireland having a slightly lower prevalence of 40% [3]. A financial, age, education, and social status gradient was also observed. These results confirmed the results of studies undertaken elsewhere, showing an association between inadequate or problematic health literacy and lower self-rated health and higher rates of chronic health conditions [3,15]. Further, the HLS-EU showed that people with inadequate and problematic health literacy and a chronic health condition found their health condition more limiting and had higher rates of healthcare utilization [3]. The HLS-EU showed associations between health literacy and self-reported physical exercise, self-reported body-mass index (BMI), and alcohol intake but no consistent association with smoking [3]. 

Whilst international data are helpful in setting the international context, it is important to undertake national-level analyses. As an example, analysis of Irish HLS-EU data confirmed the international findings of the financial, age, education, and social status gradients of health literacy [14], as well as the association between health literacy and exercise [16,17], but found no association between health literacy and BMI and alcohol intake [16]. Another difference from the international findings was a significant association between health literacy and smoking in the Irish data [16]. 

In terms of health policies, health literacy is now widely recognised as an important factor in improving health behaviours, enhancing self-management skills, improving health outcomes, addressing health inequalities, and as a lever for preventing and controlling noncommunicable diseases (NCDs). A recent evidence synthesis identified 46 existing and/or developing health literacy policies at international, national, and local levels in 19 of the 53 Member States of the World Health Organisation (WHO) European Region (36%) [18].

Ireland, like many countries across the WHO region, is actively engaged in the WHO European Action Network on Health Literacy for Prevention and Control of NCDs [19] and the Measurement of Population and Organisational Health Literacy (M-POHL) Consortium to strengthen evidence-for-policy measuring, in terms of population and organisational health literacy [20]. In terms of national health policies, strengthening health literacy in Ireland features in multiple health policies [8,21,22]. 

Like many other countries, at service level in the Irish healthcare system and in terms of health communication, there continues to be widespread recognition of the several main barriers to health literacy as outlined by Zarcadoolas et al. [23]: complexity of written information in print and on the web; lack of health information in languages other than English; lack of cultural appropriateness of health information; inaccuracy and/or incompleteness of information available through mass media; prevalence of low literacy and numeracy in the population, and among particular cohorts and groups, a lack of empowering content that targets behaviour change. Therefore, it is important to consider the extent to which, in recent years, health service providers and the national health service in Ireland (the Health Service Executive, HSE) have taken a comprehensive and multifaceted approach to overcoming the barriers to health literacy. For example, the current HSE communication strategy [24] promotes the use of plain English in all written material, champions the provision of credible online and published sources of information, supports targeted information campaigns, and messages and directs information to specific cohorts and groups, adopting behavioural insights in communication with patients and citizens [25]. In addition, the importance of adapting material to respond to known low literacy and numeracy proficiency levels in the population and particular cohorts is well recognised, and this is complimented by efforts to promote good health literacy practices in the delivery of primary and community health services, for example, through the National Adult Literacy Agency’s unique ‘Crystal Clear’ quality mark for pharmacists and general practitioners [26].

### 1.2. Health Literacy and Socio-Economic Status 

A critical motivation for this study is the evidence that low health literacy is not evenly distributed across society. There is a marked social gradient; people from lower socio-economic groups are more likely to have low health literacy [3,15,27]. Furthermore, functional health literacy may serve as a pathway by which low Socio-Economic Status (SES) affects health status [28]. An integrative review by Stormacq et al. showed that health literacy mediates the relationship between SES and health status, quality of life, specific health-related outcomes, health behaviours, and use of preventive services [29]. The authors hypothesise that health literacy can be considered as a modifiable risk factor of socioeconomic disparities in health and that enhancing the level of health literacy in the population or making health services more accessible to people with low health literacy may be a means to reach a greater equity in health [29].

There is also a well-described social gradient in chronic conditions, such as diabetes [30], cardio vascular disease [31], and chronic obstructive pulmonary disease [32], in adverse lifestyle choices [5], health care utilization [33], and self-rated health [34].

There is evidence from the UK that public health interventions have tended to have more impact in higher socio-economic groups, thus tending to widen health inequalities [5]. The presence of social gradients in health literacy, however, means that health literacy interventions may reduce rather than exacerbate health inequalities. Increasingly, effective interventions are being developed to build health literacy skills in socio-economically disadvantaged groups as well as improving healthy behaviours [35,36,37,38].

Building on this evidence, and the hypotheses put forward by Stormacq et al. [29], there are two hypotheses explored in this study:
1Increases in health literacy for people in lower social status groups are associated with a greater increase in health outcomes and healthy lifestyle behaviours than for those in middle and upper groups. 2Increases in health literacy for people in lower social status groups are associated with a greater reduction in healthcare utilisation than for those in middle and upper status groups.

The aims of this study were thus to investigate, using a population health perspective:
1The extent to which increasing health literacy may improve health outcomes and behaviours.2The potential effects of increasing health literacy at different levels of social status.

## 2. Materials and Methods

### 2.1. Data 

The data used in this study are from the European Health Literacy Survey (HLS-EU), a cross-sectional survey of adults aged 15 years and older in Austria, Germany (North Rhine-Westphalia), Greece, the Netherlands, Poland, Spain, Ireland, and Bulgaria [3]. A random sample of approximately 1000 respondents in each country was drawn and the total sample from Ireland comprised 1005 individuals. For Ireland, the survey response rate was 69%. A description of the study design and methodology for the Irish survey is available [14].

For the purpose of this study, we adopted the definition of health literacy that was developed by the HLS-EU Consortium [39]. This definition (and the associated conceptual model) was operationalized into the 47-item health literacy questionnaire: the HLS-EU-Q. In Ireland, data were collected in 2011 by the market research company TNS Opinion. 

### 2.2. Description of Measures 

There are four separate binary dependent variables of interest in this study. Smoking status was measured as yes (current smoker) or no (never smoked/former smoker). Healthcare utilisation was measured for the past 12 months and grouped into: less than 3 hospital visits, versus 3 or more hospital visits. Self-rated health status was reported as either good/very good versus fair/bad/very bad. A limiting illness was a binary variable (yes or no). The ‘yes’ category included those limited in activity by a long-term illness or health problem for at least the last 6 months. The ‘no’ category included all respondents who did not have a long-term illness and/or were not limited by their illness. This indicator was used as a proxy for disability status. 

The following covariate variables are used in this analysis. The measure of socio-economic position was perceived social status and was reported on a 10-point social ladder: one indicating the lowest level in society and 10 indicating the highest level in society. These responses were grouped into tertiles (low, middle, high) for the purpose of analysis, i.e., the lowest tertile was a self-rated score between 0 and 3.3, the middle tertile was a self-rated score between 3.4 and 6.6, and the highest tertile was a self-rated score between 6.7 and 10. Age was measured in years (range 15–91 years). Gender was measured as male or female. Educational attainment was reported using five International Standard Classification of Education (ISCED) categories, ranging from pre-primary/primary education to tertiary or higher. This analysis used the 47-item Health Literacy questionnaire with scores ranging from 11 (low) to 50 (high) self-rated health literacy. 

### 2.3. Analysis 

Probabilities were estimated using a series of logistic regression models for each SES sub-group. A product term between health literacy and SES was not introduced. Logistic regression analyses with marginal effects using the delta method were estimated to explore the extent to which health literacy is associated with the likelihood of each health status or outcome, e.g., current smoking. Models were adjusted for known determinants of health literacy and the health indicators of interest; age, gender, and education. The logit coefficients were reported as odds ratios. The significance level was set at 0.05. 

The purpose of reporting the marginal effects of the regression analysis is to observe and compare the average change in the likelihood of a health outcome based on a 5-unit change in health literacy. The delta method was used in order to obtain the appropriate standard errors (S.E.) of the smooth function of the fitted model parameter. In this analysis, we compare these results for three separate groups against the overall relationship: low, middle, and high social status groups. The purpose of this stratified approach is to determine if the change (increase in the likelihood of a good health outcome or decrease in the likelihood of a bad health outcomes) is greater for adults who are in the low social status groups compared with the middle and higher social status groups, or if the change is the same in all groups. Analyses were completed using Stata Version 14 (StataCorp LLC, College Station, TX, USA). The regression tables and models are presented in the Appendix A.

### 2.4. Ethics Approval

As this study was a secondary analysis of anonymised data, ethics approval was not required. 

## 3. Results

Table 1 shows the prevalence of the outcomes of interest in the three social status tertiles and the full sample.

Table 2 presents the sample characteristics and the mean health literacy score for each characteristic of interest.

The results of the regression analysis are presented in four groups in the following section: self-rated health status; long-standing health condition; healthcare utilisation; and smoking. The regression tables (Appendix A, Table A1, Table A3, Table A5 and Table A7), and their accompanying regression models (Appendix A, Table A2, Table A4, Table A6 and Table A8) are shown in the Appendix A. Regression models are adjusted for age, gender, and educational attainment. The results are summarized below.

### 3.1. Self-Rated Health Status

In the population as a whole and for people in all three SES tertiles, higher health literacy scores are associated with a higher probability of reporting being in ‘good or very good’ health, therefore this relationship appears to be positive and graded across the whole population. Increasing health literacy is likely to contribute to higher numbers of adults reporting being in ‘good or very good’ health in all social status groups.

### 3.2. Longstanding Health Condition 

For people in the low social status group, higher health literacy scores are associated with a lower probability of having a limiting illness. Approximately 77% of people in this social status group report having a longstanding health condition compared with 58% in the other social status groups. This pattern of association is not observed for the mid and high social status groups. Increasing health literacy is likely to contribute to lower numbers of adults with longstanding health conditions in the low social status group.

#### 3.2.1. Healthcare Utilisation: Number of Hospital Visits in the Past 12 Months

Compared with people in the low or middle social status groups, there is no association between health literacy and hospital visits among the high social status group. In the mid and low social status groups, as health literacy scores increase the probability of reporting three or more hospital visits decreases. Increasing health literacy is likely to contribute to lower numbers of hospital visits among the middle and low social status groups.

#### 3.2.2. Smoking 

For people in the low and middle social status groups, higher health literacy levels are associated with a lower probability of being a current smoker. Among the high social status group, the opposite pattern is apparent, however the effect of health literacy on smoking among this group is small and not significant. There is also a smaller number of people in the high social status group who currently smoke: 19% compared with 36% in the low social status group. Increasing health literacy is likely to contribute to lower numbers of current smokers among the middle and low social status groups. 

## 4. Discussion

There were two hypotheses underpinning this analysis. First, increasing health literacy for people in the low social status group is associated with a greater increase in health outcomes and healthy lifestyle behaviours than for those in the middle and upper social status groups. Secondly, increasing health literacy for people in the low social status group is associated with a greater reduction in healthcare utilisation than for those in middle and upper social status groups. 

People in the low social status tertile had a significantly higher prevalence of long-term chronic health conditions when compared with people from the middle and high tertiles. Furthermore, in this group, lower health literacy scores were significantly associated with the prevalence of long-term chronic health conditions, an association not seen in the middle and high social status groups. 

For smoking and health service utilisation, people in the low and middle social status tertiles had a significantly higher prevalence when compared with people from the high tertile, with a significant association between low health literacy levels and higher prevalence of smoking and higher health service utilisation in these two groups. The association between health literacy and each outcome was seen to differ within each social status group, with the exception of self-rated health status where the association was the same within each social status group. This interesting finding may reflect the fact that self-rated health status is a global measure of health that captures health and health expectations and includes social desirability bias. This is likely to be culturally sensitive. Self-rated health status is, historically, very high in Ireland; 82.8% of men and women in Ireland report good or very good health compared with 72.6% and 67.3%, respectively, across the EU 28 [6]. Evidence from the Healthy Ireland Survey has shown that year-on-year and since 2015, self-rated ’good’ health in Ireland remains high (above 70%) among current smokers, unemployed persons, and those living in deprived areas [40]. Therefore, it is not unreasonable to expect that there is not a clear association between SES and self-rated health once education, age, and gender are also controlled for. 

Although the results from the HLS-EU survey did not show any significant correlation between health literacy and smoking, an association is found in the Ireland-only data. Associations between health literacy and smoking are likely to be different between different countries, reflecting different cultures and contexts. In an international analysis (such as the HLS-EU survey), significant associations that occur in only some countries may disappear when all countries are considered together. In the Irish culture and context, smoking and health literacy are significantly associated. 

This study indicates that, for three of the four outcomes explored in this study, interventions to increase health literacy may have a larger effect for people in the lowest social status tertile (for chronic conditions) or low and middle social status tertiles (for smoking and hospital attendances) compared with people in higher tertiles. In these areas, therefore, building population health literacy may reduce health inequalities. The pattern of association between health status and health literacy at each level of social status was not statistically different. As a global measure of health and wellbeing, this is perhaps to be expected, and this result provides further evidence of the value of utilising more specific and sensitive health indicators of utilisation and behaviour to understand the impact of health literacy, as this may be overlooked when global measures alone are relied upon. 

The results of this study are consistent with previous analysis of threshold effects in health literacy among older adults in the US [41], although due to small sample sizes, it was not possible to analyse the current results by age cohort. Nevertheless, given the prevalence of these health outcomes and behaviours in all age groups presented in this study, it remains relevant to analyse the total adult population, adjusted for age. The results of this report are also concordant with previous studies that suggest that a way forward for Irish policy is to conceptualise health literacy as a top–down public health issue and consider staging regular targeted interventions for underrepresented communities who are at risk of low health literacy [1].

### Strengths and Weaknesses of the Study

The data for this analysis are high quality, collected as part of the HLS-EU 2011 [3]. The survey used a standard survey questionnaire based on a comprehensive conceptual and logic model [39], applied Eurobarometer standards [42], and ensured consistency in data collection by using one European-wide represented agency. To our knowledge, this is the first study using data to explore differences in association between health literacy and the prevalence of chronic health conditions, smoking, health service utilization, and self-rated health for different SES groups. Nevertheless, there are several limitations and avenues for further analysis that merit discussion. 

In terms of the survey data, the limitations of the HLS-EU survey are noted by Sorensen et al. [3]. The limitations pertinent to this analysis are that the sample size was restricted to 1000 respondents for each sample country. In accordance with Eurobarometer methodology, non-EU citizens living in the participating countries were not included in the survey. Indicators of personality and mental health are also not included in the study, and these are relevant factors when considering the distribution of healthcare utilisation and engagement in healthy behaviours. The measure of social status was self-reported, but the response rate was high. Alternative methods of estimating socio-economic status based on income, education, and employment were not possible, due to the absence of a variable for occupation and a high refusal rate for the question on household income (approximately 20% refusal).

In terms of the analysis presented in the study, modelling regressions and associations relies on cross-sectional data, and therefore, causality cannot be inferred. Whilst this study indicates that health literacy interventions targeted at people from lower and middle social status tertiles may reduce inequalities in the prevalence of chronic conditions, smoking, and hospital attendance, further evidence is required from the evaluation of interventions that includes measurement of SES, before and after measurement of health literacy, and sufficiently long follow-up to capture changes in the outcomes of interest. Of equal importance is that the socio-economic data in this study are self-assessed, based on a ‘ladder’ method (lowest to highest rung in society). Self-assessed socio-economic status is often used in international studies, as it enables comparison between countries, correcting for issues, such as national median incomes. Although self-assessed SES is correlated with objective SES (i.e., education, employment, and income), the correlation is only moderate (r = 0.25 to 0.34) [43]. Whilst subjective and objective SES are both associated with health, the association appears to be stronger for subjective SES [43]. If governments and their agencies seek to prioritise measures to address low health literacy in lower SES groups, it is likely that target groups for intervention would be identified through objective SES measures, i.e., education, employment, and/or income. Whilst it is reasonable to hypothesise that our findings with subjective SES would be broadly similar, were objective measures of SES to be used, further research to evaluate this would be required. 

Notwithstanding these limitations, the results of this study show that improvements in population health literacy are likely to contribute to a reduction in the prevalence of long-term chronic health conditions, reduce smoking levels, reduce hospital service utilisation, and improve self-rated health. Whilst improved health literacy should improve self-rated health across the whole population, it should have a more marked impact in lower social status groups, and hence, reduce the current social gradient in the prevalence of long-term chronic health conditions, smoking, and hospital service utilisation.

## 5. Conclusions

To conclude, it is valuable to reflect on prior analyses of the interrelationship between literacy and power as being particularly relevant to the current analysis of the social gradient in health literacy and associated consequences for health. Indeed, Rao and Rao [44] note that “… literacy comprises practices and reading and writing which enhances people’s control over their lives and their capacity for making rational judgements and decisions by enabling them to identify, understand, and act to transform social relations and practices in which power is structured unequally”. Like general literacy, strengthening health literacy is a transformative process of improving key information processing and social skills to engage with health information and health services and make good decisions about health. As such, strengthening health literacy within the population through targeted means can be considered as a mechanism for addressing socio-economic disparities in health and healthy behaviours, and the results of this study provide an empirical analysis to support this approach in Ireland. 

The recent WHO Europe evidence synthesis on health literacy policies in the region suggested that member states should “consider … developing or enhancing health literacy policies and related activities to benefit citizens, patients and communities” [18]; the findings of this paper indicate that policies and activities that focus on health literacy have the potential to achieve this, through reducing health inequalities in the prevalence of chronic health conditions, hospital utilisation, and in some countries, smoking.

## Figures and Tables

**Table 1 ijerph-17-05891-t001:** Health outcomes by self-perceived social status (low, middle, high).

	Good, Very Good Self-Perceived Health %	Longstanding Health Condition %	3 or More Hospital Visits %	Current Smoker %
All	79.68	68.25	10.72	27.70
Low	73.49	77.46	12.29	36.14
Middle	80.00	58.33	9.13	23.48
High	88.28	58.33	9.72	18.97

**Table 2 ijerph-17-05891-t002:** Sample characteristics and mean health literacy score by group.

Variable	% of Population	Mean Health Literacy Score (SD)
Total		35.17 (7.85)
**Gender**		
Male	42.89	34.34 (7.53)
Female	57.11	35.78 (8.04)
**Age**		
15–24 years	13.73	35.29 (7.67)
25–64 years	67.96	35.36 (8.03)
65–91 years	18.31	34.32 (7.27)
**Education Level**		
Levels 0,1 (Pre-primary and primary education)	7.23	33.55 (7.88)
Level 2 (Lower secondary education)	19.08	32.40 (8.24)
Level 3 (Upper secondary education)	26.20	35.27 (7.70)
Level 4 (Post-secondary non-tertiary education)	16.57	35.27 (7.26)
Levels 5,6 (First and second stage of tertiary education)	30.92	36.99 (7.61)
**Self-Perceived Social Status**		
Low	44.39	32.67 (7.79)
Medium	24.6	35.89 (6.58)
High	31.02	38.31 (7.45)
**Self-Perceived Health**		
Very bad, bad, fair	20.7	32.59 (8.16)
Good, very good	79.3	35.83 (7.64)
**Longstanding Health Condition**		
No	32.01	34.94 (7.73)
Yes	67.99	33.49 (8.26)
**Smoking Status**		
Former/never	72.44	35.59 (7.57)
Current smoker	27.56	34.04 (8.46)
**Hospital visits**		
Under 3 visits	88.62	35.37 (7.71)
3 or more visits	11.38	33.67 (8.73)

Note: SD = standard deviation.

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
