# Peer review of "Increasing Health Literacy May Reduce Health Inequalities: Evidence from a National Population Survey in Ireland"

_ijerph, 2020, doi:10.3390/ijerph17165891_

Round 1

Reviewer 1 Report

The manuscript provides an interesting insight into the association between health literacy and health outcomes in the Irish population, contributing to the growing body of knowledge on the importance of health literacy as a social determinant of health.

However, some revisions are required.

1) Title

As the authors stated in the discussion section, even if they were able to adjust for the most relevant confounders leaving no unmeasured confounding, it is not possible to draw causal inference for the association between health literacy and health outcomes, as data were retrieved through a cross-sectional design. Hence, I would change the title of the manuscript to reflect this idea.

2) Abstract

“… the association between health literacy and health status and outcomes may not be evenly distributed across society”. I would say that it is not the association to be not evenly distributed, but that it is the magnitude of the association that may vary within society.

3) Introduction

Line 57-60; the objective of the study should be stated in the last part of the introduction.

4) Methods

Line 191-193: all the primary analyses of this manuscript are based on tertiles of self-reported socioeconomic status. As tertiles, they are data-driven. Hence, it would be important to explicitly mention what are the cut-off scores used to create such tertiles.

Additionally, since this is a self-reported measure, is it possible to characterize better what it means “high SES” or “medium SES” or “low SES”? There could be a huge personal variation in how people rate/think about each level of SES (e.g., what is the income threshold to differentiate high from medium SES?).

It’s unclear how the authors estimated the probabilities for each outcome since no regression model is presented. Did they introduce a product term between health literacy and SES in each model to allow the association to varying within each SES stratum?

5) Results

Figures 2 and 3. Technically, it is not possible to have a probability less than 0: the lower bound of the confidence intervals of the estimated probabilities should be set to 0 whenever negative.

Also, the regression models were adjusted for age, sex, and education. But which value for those variables was used to estimate the reported probabilities?

I would also suggest including in the manuscript the tables with the logistic regression models instead of the estimated probabilities and their 95% confidence interval, as they are more meaningful. Tables 1A, 2A, 3A, and 4A do not add anything valuable to the paper since the same data are already presented as figures.

Lastly, the authors talk about “significant” changes in the estimated probabilities of the outcomes as a consequence of an improvement in the mean health literacy level within each SES. However, how did they test statistical significance? This part should be addressed in the methods section also.

6) Discussion

Some findings require proper discussion. For example, line 333-335: why there was no evident pattern of different associations within tertiles of SES between health literacy and self-rated health? Hypotheses?

Line 336-339. “Although the first bivariate results from the HLS EU did not show any significant correlation between health literacy and smoking, … it is perhaps not surprising this adjusted country-level analysis has demonstrated a significant association.” This sentence is unclear and seems to be contradictory.

Author Response

1) Title

As the authors stated in the discussion section, even if they were able to adjust for the most relevant confounders leaving no unmeasured confounding, it is not possible to draw causal inference for the association between health literacy and health outcomes, as data were retrieved through a cross-sectional design. Hence, I would change the title of the manuscript to reflect this idea. Thank you the title has been altered.

2) Abstract

“… the association between health literacy and health status and outcomes may not be evenly distributed across society”. I would say that it is not the association to be not evenly distributed, but that it is the magnitude of the association that may vary within society. Thank you this has been amended.

3) Introduction

Line 57-60; the objective of the study should be stated in the last part of the introduction. Thank you this has been amended.

4) Methods

Line 191-193: all the primary analyses of this manuscript are based on tertiles of self-reported socioeconomic status. As tertiles, they are data-driven. Hence, it would be important to explicitly mention what are the cut-off scores used to create such tertiles.

The cut offs were indeed data driven using a self-reported socio-economic status with a range of 0 to 10. This meant that the lowest tertile was a self-rated score between 0 and 3.3, the middle tertile was a self-rated score between 3.4 and 6.6, and the highest tertile was a self-rated score between 6.7 and 10. This has been added to the text.

Additionally, since this is a self-reported measure, is it possible to characterize better what it means “high SES” or “medium SES” or “low SES”? There could be a huge personal variation in how people rate/think about each level of SES (e.g., what is the income threshold to differentiate high from medium SES?).

Thank you for raising this important point. Self-reported measures of SES obviously differ fundamentally from objective measures such as education, employment or income.  There are several points to consider here

  1. Only subjective SES was measured in this study because it was part of an international study a subjective measure enabled comparisons between countries.
  2. Subjective and objective SES have been shown to be moderately correlated (r=0.25, 0.33 and 0.34 for education, occupation and income respectively)
  3. Both objective and subjective SES have associations with health but the association appears to be stronger for subjective SES
  4. If governments decide to invest in health literacy in low SES groups, such groups will likely be identified through objective rather than subjective SES. Whilst it is reasonable to hypothesise that our findings with subjective SES would also apply to objective SES, further research would be needed to explore this.

We have amended the text to reflect these points, and have cited this in the discussion

It’s unclear how the authors estimated the probabilities for each outcome since no regression model is presented. Did they introduce a product term between health literacy and SES in each model to allow the association to varying within each SES stratum?

Probabilities were estimated using a series of logistic regression models, for each SES sub-groups. A product term between health literacy and SES was not introduced. This is included in the text.

The regression models are now presented in the appendix. The figures have been removed from the manuscript as their content is now included in the appendix.

5) Results

Figures 2 and 3. Technically, it is not possible to have a probability less than 0: the lower bound of the confidence intervals of the estimated probabilities should be set to 0 whenever negative.

Thank you. Odds ratios are reported now in the logistic regression tables.

Also, the regression models were adjusted for age, sex, and education. But which value for those variables was used to estimate the reported probabilities?

Thank you. Details are below and are included in the appendix:

Gender: Male and female, where male was the reference category in the models.

Age was measured in years and included as a continuous variable in the models, from 15-91.

Highest level of educational attainment was measured using ISCED classifications (0- 6), and entered as a continuous variable in the models: Level 0 (Pre-primary education); Level 1 (Primary education); Level 2 (Lower secondary); Level 3 (Upper secondary); Level 4 (Post-secondary non-tertiary); Level 5 (First stage of tertiary); and, Level 6 (Second stage of tertiary).

I would also suggest including in the manuscript the tables with the logistic regression models instead of the estimated probabilities and their 95% confidence interval, as they are more meaningful. Tables 1A, 2A, 3A, and 4A do not add anything valuable to the paper since the same data are already presented as figures.

Thank you. Logistic regression models have been added to the tables in the appendix and the figures in the text have been removed

Lastly, the authors talk about “significant” changes in the estimated probabilities of the outcomes as a consequence of an improvement in the mean health literacy level within each SES. However, how did they test statistical significance? This part should be addressed in the methods section also.

The analysis is based on a logistic regression model for each outcome. The logit coefficients are reported as Odds Ratios, with p values and 95% confidence intervals are presented. This has been added to the methods and the p values are in the regression model tables (1B to 4B).

6) Discussion

Some findings require proper discussion. For example, line 333-335: why there was no evident pattern of different associations within tertiles of SES between health literacy and self-rated health? Hypotheses?

Self-rated health status is a global measure of health that captures health, health expectations and social desirability bias. This measure is culturally sensitive. Self-rated health status is, historically, very high in Ireland and remains high at different levels of objective health status and disability status, and across each SES group. Therefore, it is not unreasonable to expect that there is not clear association between SES and SR health once education, age, and gender are also controlled for. We have added text to this effect in the discussion.

Line 336-339. “Although the first bivariate results from the HLS EU did not show any significant correlation between health literacy and smoking, … it is perhaps not surprising this adjusted country-level analysis has demonstrated a significant association.” This sentence is unclear and seems to be contradictory.

The point we are making here is that associations between health literacy and smoking are likely to be different between different countries, reflecting different cultures and contexts. In an international analysis (such as the HLS EU), significant associations that occur in the minority of countries may disappear when all countries are considered together. We have clarified this in the text.

Reviewer 2 Report

I am writing about manuscript entitled 'The impact of increasing health literacy on health inequalities: evidence from a national population survey in Ireland'. The manuscript investigates the associations between health literacy and health status and outcomes. This study aims to estimate and compare the associations among health status, health behaviors, and healthcare utilization within different levels of social status in the Irish population. In addition, Logistic regression analysis was conducted to estimate the likelihood of each health outcome. Marginal effects were calculated using the delta method to demonstrate the change in likelihood of each outcome associated with a 5-point increase in health literacy score and reveals that (a) Higher health literacy scores were associated with a lower probability of having a limiting illness within the low social status group (b) Increasing health literacy is likely to contribute to lower numbers of hospital visits among the middle and low social status groups.(c) For people in the low and middle social status groups, higher health literacy levels were associated with a lower probability of being a current smoker.(d)The associations between health literacy and self-rated health status were similar in each social status group.

A-     Revisions suggested: Minor

1-     Author might revise below statements/sentences:

(1). Abstract

 Higher health literacy scores were associated with a lower probability of 3 more hospital visits in the past 12 months in the low and middle social status groups.

(2). Introduction

Mental health is also an issue as there are higher rates of depression and hospitalization among lower socio-economic classes in Ireland, often due to the mental health effects of poverty, systemic inequality and material deprivation.

 Evidence from a European survey “Health Inequalities in Europe: Setting The Stage for Progressive Policy Action” [10] suggests that Ireland’s two-tiered healthcare system may also contribute to the gap of health inequity insofar as a proportion of the population on low incomes are above the income threshold of entitlement for the General Medical Scheme (GMS), pay out-of-pocket for medical expenses, and cannot afford private medical insurance.

It is clear that the health and social care system in Ireland is complex and like many countries also presents a particular challenge in terms of health literacy and public health, insofar as those most in need are also those who are likely to lack the health literacy skills to navigate the system and engage fully in programs and interventions to improve health and wellbeing.

(3). Longstanding health condition

Approximately 77% of people in this social status group report having a longstanding health condition compared with 58% in the other social status groups.

    (4). Healthcare utilization: number of hospital visits in the past 12 months

 As health literacy scores increase the probability of reporting 3 or more hospital visits decreases.

(5). smoking

Higher health literacy levels are associated with a lower probability of being a current smoker.

 There are also a smaller number of people in the high social status group who currently smoke.

(6).discussion

The pattern of differential associations between health literacy and health outcomes for different social status groups was not seen in self-rated health, where the association was observed across all groups.

B: Revisions suggested: Major

  1. Author concludes associations between health status, health behaviors, and healthcare utilization within different levels of social status in the Irish population. But the literature review is poor, and then it is strongly suggested that author should add some more literature on Health literacy and socio-economic status.

  1. The criterion of references should be careful. Such as: Evaluation of the second phase of the Skilled for Health Programme; The Tavistock Institute and Shared Intelligence: London, 2009.

  1. Authors should highlight the sufficient contributions to the new body of knowledge and policy.

  1. More recommendations for the decision makers are expected in the conclusions.

  1. Author need pay attention to punctuation in the paper. For example: for all adults and for three separate groups; low, medium, and high social status.

Author Response

A-     Revisions suggested: Minor

1-     Author might revise below statements/sentences:

(1). Abstract

 Higher health literacy scores were associated with a lower probability of 3 more hospital visits in the past 12 months in the low and middle social status groups. Thank you. Corrected.

(2). Introduction

Mental health is also an issue as there are higher rates of depression and hospitalization among lower socio-economic classes in Ireland, often due to the mental health effects of poverty, systemic inequality and material deprivation. Thank you. Amended.

Evidence from a European survey “Health Inequalities in Europe: Setting The Stage for Progressive Policy Action” [10] suggests that Ireland’s two-tiered healthcare system may also contribute to the gap of health inequity insofar as a proportion of the population on low incomes are above the income threshold of entitlement for the General Medical Scheme (GMS), pay out-of-pocket for medical expenses, and cannot afford private medical insurance. Thank you. Amended.

It is clear that the health and social care system in Ireland is complex and like many countries also presents a particular challenge in terms of health literacy and public health, insofar as those most in need are also those who are likely to lack the health literacy skills to navigate the system and engage fully in programs and interventions to improve health and wellbeing. The sentence has been shortened and clarified.

(3). Longstanding health condition

Approximately 77% of people in this social status group report having a longstanding health condition compared with 58% in the other social status groups.

We feel that this sentence is correct, we have not altered it.

    (4). Healthcare utilization: number of hospital visits in the past 12 months

 As health literacy scores increase the probability of reporting 3 or more hospital visits decreases.

We feel that this sentence is correct, we have not altered it.

(5). smoking

Higher health literacy levels are associated with a lower probability of being a current smoker.

Thank you, this part of the sentence has been replaced as suggested below i.e.’There are also a smaller number of people in the high social status group who currently smoke.’

(6).discussion

The pattern of differential associations between health literacy and health outcomes for different social status groups was not seen in self-rated health, where the association was observed across all groups.

Thank you we have rephrased this to make it clearer. It now reads ‘The association between health literacy and each outcome was seen to differ within each social status group, with the exception of self-rated health status where the association was the same within each social status group.’

B: Revisions suggested: Major

  1. Author concludes associations between health status, health behaviors, and healthcare utilization within different levels of social status in the Irish population. But the literature review is poor, and then it is strongly suggested that author should add some more literature on Health literacy and socio-economic status.

Thank you for this insightful comment. We have amended this section in the introduction in the following ways.

  1. We have added in a reference from this year confirming the SES gradient in health literacy (Svendsen et al). 
  2. We have added the evidence from Lastrucci et al. indicating that functional HL may serve as a pathway by which SES affects health status, especially in lower SES groups.
  3. We have added in the work by Stormacq et al, whose integrative review showed that health literacy mediates the relationship between SES and health status, quality of life, specific health-related outcomes, health behaviours and use of preventive services. They posit that health literacy can be considered as a modifiable risk factor of socioeconomic disparities in health. Enhancing the level of HL in the population or making health services more accessible to people with low HL may be a means to reach a greater equity in health.

  1. The criterion of references should be careful. Such as: Evaluation of the second phase of the Skilled for Health Programme; The Tavistock Institute and Shared Intelligence: London, 2009.

Thank you we have amended this

  1. Authors should highlight the sufficient contributions to the new body of knowledge and policy.

Thank you. We have added some information within the introduction section about current knowledge of the extent of health literacy policies in Europe and added some sentences about the policy implications of this work at the end of the Conclusion.

  1. More recommendations for the decision makers are expected in the conclusions.

Thank you we feel this is addressed through the additional text outlined in the response above.

  1. Author need pay attention to punctuation in the paper. For example: for all adults and for three separate groups; low, medium, and high social status.

This has been reviewed and amended where necessary.

Reviewer 3 Report

  1. In the introduction it is too much about Ireland as again a large part of article is devoted to Ireland and health literacy.
  2. Also, the goal and hypothesis should appear earlier . Then the reading would be more clear. 
  3. There is not so much information about some others research in this field (analysis of previous research some other research - what kind of methods were used etc). There is only one comparison (line 350) 

Author Response

  1. In the introduction it is too much about Ireland as again a large part of article is devoted to Ireland and health literacy.

We respectfully highlight to the reviewer that this is an Irish study undertaken exclusively with Irish data. Where possible we have illustrated the international background work and implications of our findings.

  1. Also, the goal and hypothesis should appear earlier . Then the reading would be more clear. 

Perhaps state near the beginning of the Introduction:

Thank you. We have added text earlier in the introduction section to clarify this.

  1. There is not so much information about some others research in this field (analysis of previous research some other research - what kind of methods were used etc). There is only one comparison (line 350) 

Thank you. We have added the following additional information:

  1. The importance of objective classification vs subjective rating of socio-economic status and it’s implications in this study (Cundiff et al).
  2. We have added in a reference from this year confirming the SES gradient in health literacy (Svendsen et al) .
  3. We have added the evidence from Lastrucci et al. indicating that functional HL may serve as a pathway by which SES affects health status, especially in lower SES groups.
  4. We have added in the work by Stormacq et al, whose integrative review showed that health literacy mediates the relationship between SES and health status, quality of life, specific health-related outcomes, health behaviours and use of preventive services. They posit that health literacy can be considered as a modifiable risk factor of socioeconomic disparities in health. Enhancing the level of HL in the population or making health services more accessible to people with low HL may be a means to reach a greater equity in health.

Round 2

Reviewer 1 Report

The authors addressed all reviewers' comments and suggestions.

Reviewer 2 Report

I agree with the revision of authors.

Reviewer 3 Report

The comment were taken into account.